# Identification of risk factors and development of a predictive model in patients using cefmetazole for international normalized ratio elevation

Takaya Namiki[1,2], Yuta Yokoyama [1,3]*, Motonori Kimura[4], Shogo Fukuda[4], Shoji Seyama[5¤], Osamu Iketani[6], Masaru Samura[7], Haruki Ishikawa[4], Aya Jibiki[3], Hitoshi Kawazoe[1,3], Hisakazu Ohtani[4], Naoki Hasegawa[8], Kazuaki Matsumoto[9], Hideki Hashi[2], Sayo Suzuki [1,3], Tomonori Nakamura[1,3]

**1** Division of Pharmaceutical Care Sciences, Keio University Graduate School of Pharmaceutical Sciences, Tokyo, Japan, **2** Department of Pharmacy, Tokyo Bay Urayasu Ichikawa Medical Center, Chiba, Japan, **3** Division of Pharmaceutical Care Sciences, Center for Social Pharmacy and Pharmaceutical Care Sciences, Keio University Faculty of Pharmacy, Tokyo, Japan, **4** Department of Pharmacy, Keio University Hospital, Tokyo, Japan, **5** Division of Infectious Diseases and Infection Control, Keio University Hospital, Tokyo, Japan, **6** Division of Academic Research Support, Keio University Hospital, Tokyo, Japan, **7** Department of Pharmacy, Yokohama General Hospital, Kanagawa, Japan, **8** Department of Infectious Diseases, Keio University School of Medicine, Tokyo, Japan, **9** Division of Pharmacodynamics, Keio University Faculty of Pharmacy, Tokyo, Japan

¤ Current address: Department of Clinical Microbiology, School of Pharmacy, Tokyo University of Pharmacy and Life Sciences, Tokyo, Japan
* yokoyama-yt@pha.keio.ac.jp

## Abstract

Patient risk factors related to coagulopathy and bleeding when using cefmetazole (CMZ) have not yet been identified, and no models exist to predict side effects during CMZ treatment. Moreover, reports that examine which patients should be careful when using CMZ to ensure safety are lacking. Our objective was to understand risk factors for elevated international normalized ratio (INR) in patients using CMZ and to develop a predictive model for INR elevation using a risk score to enable safe administration of CMZ. This multicenter, retrospective, and observational study was conducted in Tokyo Bay Urayasu Ichikawa Medical Center and Keio University Hospital using data from patients being treated with CMZ. Patients were classified into INR-elevated or non-INR-elevated groups. Univariate and multivariate analyses were performed to calculate the adjusted odds ratios (aOR) and 95% confidence intervals (CI). The actual probability of an elevated INR and probability of an elevated INR predicted by the regression β coefficients were calculated and classified into four categories according to the risk score. Binomial logistic regression analysis revealed that liver disorder (aOR, 5.65; 95% CI, 1.69–18.91; risk scores, 2), nutritional risk (aOR, 6.32; 95% CI, 3.14–12.74; risk scores, 2), no-diabetes mellitus (aOR, 4.53; 95% CI, 1.34–15.26; risk scores, 2), and warfarin use (aOR, 98.44; 95% CI, 7.05–1375.50; risk scores, 5) were significantly associated with INR elevation. The predicted incidence probabilities of INR elevation were < 5% (low risk), 5–< 30% (medium risk),

**Data availability statement:** We have made the data available in a public repository on Figshare. You can access the data through the following private link URL: https://figshare.com/s/a7cc966004fa34980f25.

**Funding:** The author(s) received no specific funding for this work.

**Competing interests:** Dr. H. K. received research funding from Eli Lilly outside of the submitted work. Dr. K. M. received grants from Meiji Seika Pharma, Sumitomo Pharma and Shionogi outside of the submitted work. T.N. received research funding from Chugai, Daiichi Sankyo, Kyowa Kirin, Otsuka Pharmaceutical, Sanofi, and Shionogi outside of the submitted work. Conflicts that the editors consider relevant to the manuscript content have been disclosed. There are no conflicts of interest to declare. Although Dr. H. K, Dr. K. M, and Dr. T. N have received funding outside of the research, this does not alter our adherence to PLOS ONE policies on sharing data and materials. We have submitted to ICMJE.

30–<90% (high risk), and ≥ 90% (very high risk). The model validity showed a good fit (AUC, 0.79; 95% CI, 0.73–0.85, P<0.001). We identified risk factors that contribute to INR elevation and constructed a model to predict INR elevation using the risk score. Using this predictive model enables the appropriate use of CMZ in a safe manner.

## Introduction

Extended-spectrum β-lactamase-producing enterobacteria (ESBL-E) have become a major public health challenge owing to increasing number of patients affected by antimicrobial-resistant bacteria between 2012 and 2017, leading to increased morbidity and mortality rates [1,2]. ESBL-E strains are usually treated with carbapenems [3], but the emergence of carbapenem-resistant Enterobacteriaceae due to increased use of carbapenems remains a point of concern [4].

Among the cephem antibiotics, cefmetazole (CMZ), which is a cephamycin antibiotic, shows stability against hydrolysis of ESBL-E [5] and has recently attracted attention as an alternative to carbapenems [6,7]. The only cephamycin-type antibiotic available in Japan is cefmetazole, and cefotetan (CTT), which is available in the United States, cannot be used. However, CMZ and CTT has an *N*-methyltetrazolethiol (NMTT) group in its structure, and it has been reported that the NMTT group inhibits vitamin K 2, 3-epoxide reductase (VKOR) [8]. VKOR is required to convert vitamin K epoxide to vitamin K [9,10]. Inhibition of VKOR inhibits the vitamin K cycle and prevents the production of vitamin K. Therefore, it has been reported that the use of antibiotics containing NMTT groups is significantly associated with coagulopathy and bleeding [11–13]. Risk factors have been reported for some antibiotics containing the NMTT group. Risk factors for coagulopathy reported for latamoxef include cumulative defined daily doses (DDDs) and nutritional risk [11]. Risk factors for bleeding reported for cephem antibiotics include anticoagulant use, liver disorder, nutritional risk, and history of hemorrhagic events 6 months prior to the index date. Risk factors for bleeding for CMZ have not yet been identified, nor is there a model for predicting bleeding side effects for CMZ [12]. Furthermore, there are no reports that identified risk factors such as area under the blood concentration-time curve (AUC), which is an index of internal exposure, by calculated pharmacokinetic parameters for each patient treated with CMZ. In addition, there are no reports examining what kind of patients require caution when using CMZ in order to ensure safety.

This study aimed to identify the risk factors for INR elevation due to CMZ use, including determining AUC, and to construct a model to predict INR elevation using risk scores, thereby enabling the safe administration of CMZ.

## Materials and methods

### Patient information

This retrospective observational study was conducted on patients who were being administered CMZ at Tokyo Bay Urayasu Ichikawa Medical Center and

Keio University Hospital between January 2019 and October 2022. This study was reviewed and approved by the central comprehensive review of the Keio University School of Medicine Ethics Committee (Approval date: January 16, 2023, approval Number: 20221159). Approval was also obtained from the Tokyo Bay Urayasu Ichikawa Medical Center Ethics Review Committee (Approval date: January 27, 2023, approval Number: 819) and the Research Ethics Committee of the Faculty of Pharmaceutical Sciences, Keio University (Approval date: May 30, 2023, approval Number: 230530−1). Informed consent was obtained using an opt-out document on the hospital website. This study did not include minors. Data was collected in a form that does not allow personal information to be identified. The dates of access to electronic medical record data were from June/1/2023 to July/31/2023, after obtaining approval from the ethics committee of each facility.

Patients for whom INR was unmeasured prior to commencing CMZ use, patients who were prescribed CMZ for prevention of surgical site infections, and patients with a history of CMZ use were excluded. Next, mainly when starting CMZ, patients were excluded according to the following criteria: continuous bleeding from before CMZ use, unstable baseline INR (prolonged from baseline INR due to acute illness, e.g., septic shock), started concurrently with other antibiotics, intravenous administration of vitamin K preparation during the hospitalization period, unstable effect of warfarin and direct oral anticoagulants (DOACs), continuous hemodiafiltration patients, intermittent hemodiafiltration patients, pregnant women, and children (Under 15 years old).

## Data collection

Drug and molecular target nomenclature conforms to the International Union of Basic and Clinical Pharmacology/British Pharmacological Society (IUPHAR/BPS) Guide standards [14]. The following patient information was collected from electronic medical records on the INR maximum day during CMZ use: age, sex, body weight, body mass index (BMI), creatinine clearance ($CL_{CR}$) using the Cockcroft-Gault formula, albumin (ALB), D-dimer, baseline INR (within 7 days prior to starting CMZ use), maximum INR when using CMZ, dosage up to the day of maximum INR, daily dose, AUC, hypertension, no-diabetes mellitus (patients not diagnosed with diabetes mellitus), cancer, nutritional risk (NRS2002 score of 3 or higher), [15] liver disorder (Child-Pugh classification of B or higher), hemodialysis, antibiotic use history within the past 6 months, surgery history within the past 6 months, bleeding history within the past 6 months, type of infection, concomitant use of warfarin, DOACs, antiplatelet drugs, nutrient drugs (containing vitamin K), NSAIDs, and steroids. Elevated INR was defined as a ratio of maximum INR to baseline INR (maximum INR divided with baseline INR) during CMZ use greater than 1.25 [11]. The NRS2002 score consists of four initial screening items, and if any of these items apply, final screening is carried out. 1: BMI is less than 20.5, 2: The patient has lost weight within the past 3 months, 3: The patient has decreased food intake in the past week, 4: The patient is seriously injured (e.g., is the patient receiving treatment in an Intensive Care Unit?) At the final screening, participants were scored according to the severity of malnutrition and the severity of disease or trauma: None, score 0; mild, score 1; moderate, score 2; severe, score 3. If patient age is 70 years or greater, add 1 to each total [15]. The dosage was evaluated as a daily dose, taking into consideration the difference due to the time of blood sampling. AUC was calculated by dividing the dose on the day of the highest INR by unbound CMZ clearance (CL). We performed a population pharmacokinetic analysis using unbound CMZ concentration in non-hemodialysis and constructed a final model formula of unbound CMZ. The final model formula was CL = 16.2 × $(CL_{CR}/4.36)^{0.781}$ × $(ALB/28)^{1.2}$ L/h [16]. The CL of unbound CMZ was calculated by substituting $CL_{CR}$ and ALB for each patient into this formula. For CL in patients undergoing hemodialysis, we used the mean values of non-hemodialysis CL ($CL_{non-HD}$) and hemodialysis CL ($CL_{HD}$) that we clarified. $CL_{non-HD}$ and $CL_{HD}$ were 0.803 L/h and 12.6 L/h [17]. AUC was calculated by dividing the dose for each patient by the sum of $CL_{HD} + CL_{non-HD}$ according to the hemodialysis time and the non-hemodialysis time. To define as stable state of warfarin therapy, we confirmed the following: no dose changes for at least 7 days prior to CMZ use and no dose change during CMZ use, 2 or more measurements of INR within the

therapeutic range during the 7 days prior to commencement of CMZ use [18–20]. DOACs were patients with no dose changes for during and within 7 days prior to CMZ use.

## Statistical analysis

SPSS version 28.0 (SPSS Inc., Chicago, IL) and R (Version 4.2.1) were used for data analysis. The following factors were compared between groups to identify risk factors for INR elevation when using CMZ; age, sex, body weight, BMI, $CL_{CR}$, ALB, baseline INR, daily dose, AUC, hypertension, no-diabetes mellitus, cancer, nutritional risk, liver disorder, hemodialysis, antibiotic use history within the past 6 months, surgery history within the past 6 months, bleeding history within the past 6 months, type of infection, concomitant use of warfarin, DOACs, antiplatelet drugs, nutrients drugs, NSAIDs, and steroids. Multicollinearity confirmation retained factors with lower P-values during uni-variate analysis when the correlation coefficient between each factor > 0.4. Furthermore, the variance inflation factor (VIF) was also confirmed. A VIF of 5 or more suggested the possibility of multicollinearity, and a VIF of 10 or more was multicollinear. Continuous variables were subjected to the Mann–Whitney $U$ test after confirmation of normality. Fisher's exact test and chi-square test were performed for categorical variables. Multivariate analysis was performed using the forced entry method for factors with $P < 0.1$ as a result of univariate analysis. To change the cut-off value, receiver operating characteristic (ROC) analysis was performed, the optimum value was obtained, and multivariate analysis was performed again. From the results of multivariate analysis using the stepwise method with $P < 0.05$ as statistically significant, risk score assignment and INR elevated prediction probability were determined. Multivariate analysis was used to calculate adjusted odds ratios (aOR) and 95% confidence intervals (CI) to identify risk factors. Based on previous reports, we assigned a risk score to each risk factor using the β coefficient obtained from the binomial logistic regression analysis [21,22]. From the total score of each risk factor, four risk classifications of low risk (< 5%), medium risk (5–< 30%), high risk (30–< 90%), and very high risk (≥ 90%) were performed for each elevated INR occurrence probability [21,23]. The adequacy of the risk prediction model for the probability of elevated INR associated with CMZ was assessed using the Hosmer–Lemeshow test, and predictive and observed estimates of elevated INR associated with CMZ were compared using calibration plots. A bootstrap of 1000 replicates was set and a calibration curve was calculated using regression analysis. Verification was also performed using an ROC curve that evaluated the AUC.

## Results

### Patient data

The study covered 4,877 patients who used CMZ, with 565 intermediate target patients, for a final total of 429 patients after applying the exclusion criteria (61 in the elevated group and 368 in the non-elevated group) (Fig 1). The median age of the final target patients was 72 (interquartile range [IQR]: 60–82), and 74 (IQR: 65–83) and 72 (IQR: 60–82) years in the INR-elevated and non-INR-elevated groups, respectively. The proportion of males among the final target patients was 57.3% in the final target patients and 59% and 57.1%, in the INR-elevated and non-INR elevated groups, respectively (Table 1).

### INR-elevated risk factor identification

Univariate analysis showed $P < 0.1$ and statistically significant difference in body weight, BMI, $CL_{CR}$, ALB, AUC, no-diabetes mellitus, nutritional risk, liver disorder, history of surgery within the last 6 months, and concomitant use of warfarin and DOACs. Correlations were confirmed and correlations were shown between body weight and BMI (r = 0.86, $P < 0.001$) and $CL_{CR}$ (r = 0.424, $P < 0.001$), respectively. According to univariate analysis, body weight was excluded because the P values for BMI and $CL_{CR}$ were lower than the P values for body weight. AUC was correlated with various

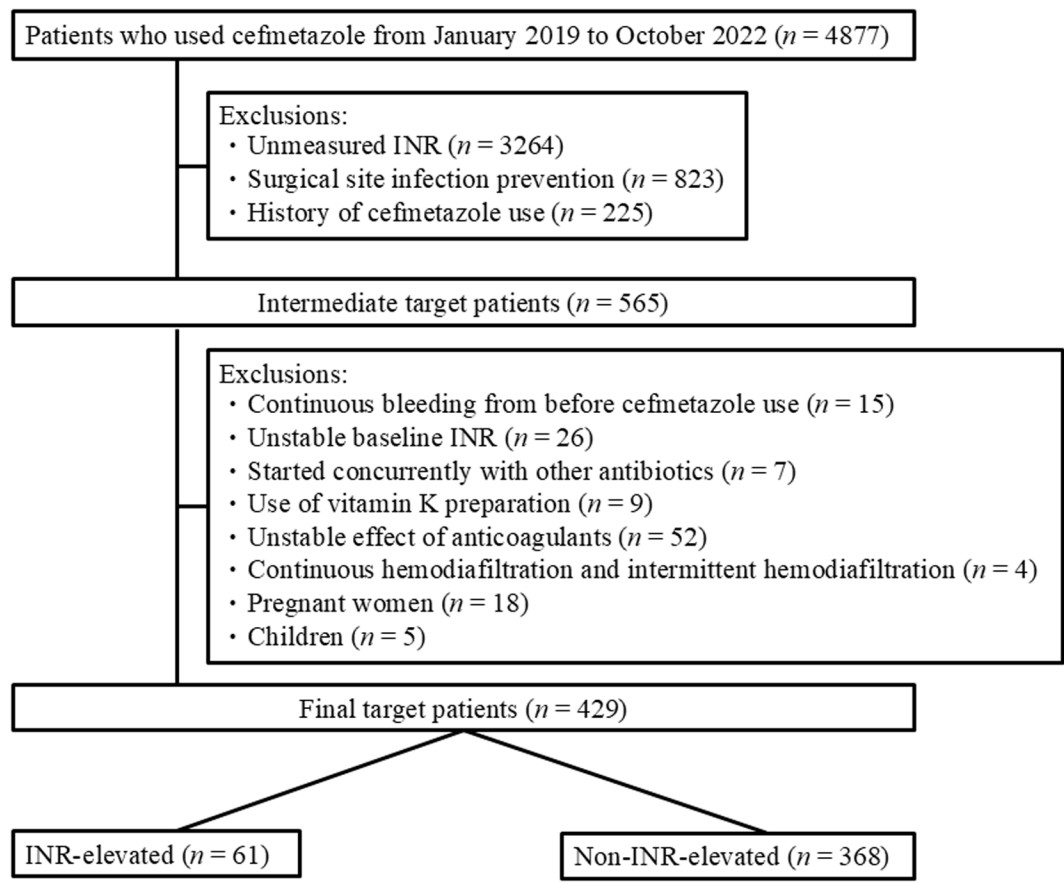

**Fig 1. Flowchart describing study cohort enrollment, international normalized ratio (INR)-elevated group, and non-INR-elevated group.**

factors including ALB (r = 0.390, P < 0.001), hemodialysis (r = 0.287, P < 0.001), $CL_{CR}$ (r = 0.240, P < 0.001), warfarin (r = 0.213, P < 0.001), and nutritional risk (r = 0.205, P < 0.001), respectively. The VIF results showed VIF < 5, indicating no multicollinearity. Binomial logistic regression analysis revealed liver disorder (aOR, 5.65; 95% CI, 1.69–18.91), nutritional risk (aOR, 6.32; 95% CI, 3.14–12.74), no-diabetes mellitus (aOR, 4.53; 95% CI, 1.34–15.26), and warfarin (aOR, 98.44; 95% CI, 7.05–1375.50) were significant at P < 0.05. The predicted correctness probability of the model was 73.8% sensitivity, 75.5% specificity, and 75.3% overall with the change in the cut-off value of binomial logistic regression analysis using ROC analysis. Although AUC was statistically significant at P = 0.04, it did not contribute to the elevated INR (aOR, 1.00; 95% CI, 1.00–1.00).

### INR-elevated prediction model construction and accuracy verification

The results of stepwise binomial logistic regression analysis based on liver disorder, nutritional risk, no-diabetes mellitus, and use of warfarin are shown in Table 2. Using the β coefficient, risk scores were assigned as follows: liver disorder, 2; nutritional risk, 2; no-diabetes mellitus, 2; warfarin, 5. A risk score was calculated for each patient, and the predicted occurrence probability obtained from the regression equation and the actual occurrence probability for each risk score are shown in Fig 2. Risk scores between 0 and 2 were low risk (predicted probability < 5%), a risk score of 4 was medium risk (predicted probability 5–< 30%), those between 6–7 were high risk (predicted probability 30–< 90%),

**Table 1. Final target patients of this study and Univariate and multivariate analysis of risk factors for cefmetazole (CMZ)-induced international normalized ratio (INR) elevation.**

| | Final target patients (n=429) | Univariate analysis | | | Multivariate analysis | |
|---|---|---|---|---|---|---|
| | | INR-elevated (n=61) | Non-INR-elevated (n=368) | *P* value | Adjusted OR (95% CI) | *P* value |
| Age (years old) | 72 (60–82) | 74 (65–83) | 72 (60–82) | 0.364 | | |
| Sex (male) | 246 (57.3) | 36 (59.0) | 210 (57.1) | 0.775 | | |
| Body weight (kg) | 56.2 (47.8–66.0) | 52.7 (45.2–63.1) | 56.9 (48.5–66.2) | 0.063 | | |
| BMI (kg/m$^2$) | 21.6 (19.3–24.5) | 20.3 (18.4–23.1) | 21.8 (19.6–24.6) | 0.016 | 1.02 (0.94–1.11) | 0.645 |
| CL$_{CR}$ (mL/min)[a] | 64.8 (42.5–91.7) | 51.8 (35.9–78.7) | 66.9 (43.7–93.2) | 0.011 | 1.00 (0.99–1.01) | 0.596 |
| Albumin (g/L) | 30 (25–34) | 27 (23–31) | 30 (25–35) | 0.008 | 1.64 (0.93–2.88) | 0.086 |
| D-dimer[b] | 5.8 (2.75–10.65) | 7.39 (3.9–11.43) | 5.6 (27–9.85) | 0.378 | | |
| Baseline INR | 1.05 (0.99–1.14) | 1.05 (0.97–1.16) | 1.05 (0.99–1.14) | 0.864 | | |
| Total dose (g)[c] | 3 (2–8) | 6 (2–13) | 3 (2–7) | 0.025 | | |
| 4 g or less daily | 401 (93.5) | 57 (93.4) | 344 (93.5) | 0.992 | | |
| Over 4 g daily | 28 (6.5) | 4 (6.6) | 24 (6.5) | 0.992 | | |
| AUC (g· h/L)[d] | 258.4 (100–697.2) | 585.7 (135–1597) | 245.1 (97.5–582.4) | <0.001 | | |
| Hypertension | 229 (53.4) | 34 (55.7) | 195 (53.0) | 0.690 | | |
| No-diabetes mellitus | 349 (81.4) | 55 (90.2) | 294 (79.9) | 0.056 | 4.53 (1.34–15.3) | 0.015 |
| Cancer | 144 (33.6) | 22 (36.1) | 122 (33.2) | 0.655 | | |
| Nutritional risk[e] | 194 (45.2) | 49 (80.3) | 145 (39.4) | <0.001 | 6.32 (3.14–12.7) | <0.001 |
| Liver disorder[f] | 338 (78.8) | 58 (95.1) | 280 (76.1) | 0.001 | 5.65 (1.69–18.9) | 0.005 |
| Hemodialysis | 14 (3.3) | 5 (8.2) | 9 (2.4) | 0.036 | 3.08 (0.58–16.3) | 0.185 |
| Antibiotic use history[g] | 194 (45.2) | 32 (52.5) | 162 (44.0) | 0.22 | | |
| Surgical history[g] | 16 (3.7) | 5 (8.2) | 11 (3.0) | 0.062 | 1.55 (0.39–6.09) | 0.533 |
| Bleeding history[g] | 61 (14.2) | 7 (11.5) | 54 (14.7) | 0.437 | | |
| Type of infection | | | | | | |
| Urinary tract | 44 (10.3) | 3 (4.90) | 41 (11.1) | 0.138 | | |
| Gastrointestinal disease | 347 (80.9) | 50 (82.0) | 297 (80.7) | 0.817 | | |
| Pneumonia | 29 (6.8) | 6 (9.8) | 23 (6.3) | 0.278 | | |
| Bacteremia | 48 (11.2) | 5 (8.2) | 43 (11.7) | 0.423 | | |
| Others[h] | 0 (0) | 10 (2.3) | 10 (2.7) | 0.37 | | |
| Concomitant use | | | | | | |
| Warfarin | 6 (1.4) | 5 (8.2) | 1 (0.3) | <0.001 | 98.4 (7.05–1375.5) | <0.001 |
| DOACs | 24 (5.6) | 7 (11.5) | 17 (4.6) | 0.040 | 1.75 (0.54–5.64) | 0.347 |
| Antiplatelet drugs | 83 (19.3) | 13 (21.3) | 70 (19.0) | 0.675 | | |
| Nutrient drugs[i] | 28 (6.5) | 5 (8.2) | 23 (6.3) | 0.575 | | |
| NSAIDs | 202 (47.1) | 29 (47.5) | 173 (47.0) | 0.939 | | |
| Steroids | 74 (17.2) | 9 (14.8) | 65 (17.7) | 0.578 | | |

Data are presented as the median [interquartile range] for continuous variables and numbers (%) for categorical variables.

Abbreviations: BMI, body mass index; AUC, area under the curve; DOACs, direct oral anticoagulant; INR, international normalized ratio; NSAIDs, non-steroidal anti-inflammatory drugs.

[a]CL$_{CR}$ was estimated using the Cockcroft-Gault equation with measured body weight.

[b]Data from 65 people who measured D-dimer.

[c]Total dose was dosage up to the day of maximum INR.

[d] AUC was calculated by dividing the total dose up to the day of maximum INR by unbound CMZ clearance (CL). For non-hemodialysis patients, the CL was calculated from CL$_{CR}$ and albumin for each patient. For hemodialysis patients, the CL of hemodialysis and non-hemodialysis were calculated according to the hemodialysis time and the non-hemodialysis time.

*(Continued)*

**Table 1.** (Continued)

[e]Nutritional risk was NRS2002 score of 3 or higher.

[f]Liver disorder was Child-Pugh classification of B or higher.

[g]Within the last 6 months.

[h]Others were patients with skin infections, bone infections, renal abscesses, and febrile neutropenia.

[i]Nutrient drugs refers to drugs containing vitamin K.

**Table 2. Multivariable analysis of risk factors for cefmetazole-induced international normalized ratio (INR) elevation.**

| Risk factor | β | SE | Wald | p value | Adjusted OR (95% CI) | Risk score |
|---|---|---|---|---|---|---|
| Liver disorder[a] | 1.73 | 0.62 | 7.88 | 0.005 | 5.65 (1.69–18.9) | 2 |
| Nutritional risk[b] | 1.84 | 0.36 | 26.6 | <0.001 | 6.32 (3.14–12.7) | 2 |
| No-diabetes mellitus | 1.51 | 0.62 | 5.93 | 0.015 | 4.53 (1.34–15.3) | 2 |
| Warfarin | 4.59 | 1.35 | 11.6 | <0.001 | 98.4 (7.05–1375.5) | 5 |
| Intercept | −5.86 | 0.88 | 44.0 | <0.001 | 0.003 | |

[a] Liver disorder was Child-Pugh classification of B or higher.

[b] Nutritional risk was NRS2002 score of 3 or higher.

Abbreviations: β, partial regression coefficient; SE, standard error; OR, odds ratio; CI, confidence interval.

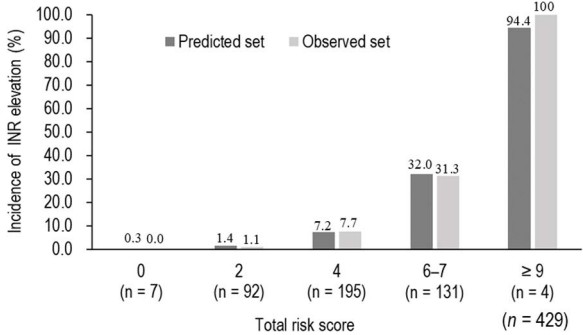

**Fig 2. Incidence of international normalized ratio (INR) elevation for increasing risk score in the predicted set and the observed set with patients using CMZ.** The individual risk score was calculated using Table 2. The percentages of incidence of INR elevation in the prediction set with patients using CMZ are shown as the mean of each score.

and those of 9 or higher were very high risk (predicted probability of occurrence ≥ 90%). The probability of INR elevation for each risk factor possessed by patients using CMZ is shown in Table 3. Patients with all risk factors had a risk score of 11 (very high risk). Patients with two of the three risk factors for liver disorder, nutritional risk, and no-diabetes mellitus, in addition to warfarin use, had a risk score of 9 (very high risk). Patients with all risk factors (liver disorder, nutritional risk, and no-diabetes mellitus), except for warfarin use, had a risk score of 6 (high risk). Patients not using warfarin and with one of the three risk factors, namely, liver disorder, nutritional risk, and no-diabetes mellitus, had a risk score of 2 (low risk).

The Hosmer–Lemeshow test showed $P = 0.782$, and the calibration plot showed a small mean absolute error of 0.004, demonstrating the validity of the model (Fig 3). In addition, the validity of the model showed good fit in the ROC analysis (AUC, 0.79; 95% CI, 0.73–0.85, $P < 0.001$) (Fig 4).

Table 3. Probability of cefmetazole-induced international normalized ratio (INR) elevation for each risk factor possessed by the patient.

| Risk classification | Incidence of INR elevation | Risk factor | | | |
|---|---|---|---|---|---|
| | | Liver disorder | Nutritional risk | No-diabetes mellitus | Warfarin |
| Very high risk | ≥ 90% | Yes | Yes | Yes | Yes |
| | | Yes | Yes | No | Yes |
| | | No | Yes | Yes | Yes |
| | | Yes | No | Yes | Yes |
| High risk | 30–<90% | Yes | No | No | Yes |
| | | No | Yes | No | Yes |
| | | No | No | Yes | Yes |
| | | Yes | Yes | Yes | No |
| Moderate risk | 5–<30% | Yes | Yes | No | No |
| | | No | Yes | Yes | No |
| | | Yes | No | Yes | No |
| Low risk | < 5% | Yes | No | No | No |
| | | No | Yes | No | No |
| | | No | No | Yes | No |
| | | No | No | No | No |

[a]Liver disorder was Child-Pugh classification of B or higher.

[b]Nutritional risk was NRS2002 score of 3 or higher.

## Bleeding of patients using CMZ

Bleeding of patients using CMZ was observed in 3 patients in the INR-elevated group (4.9%) and 3 patients in the non-INR-elevated group (0.82%), $P = 0.04$, showing a statistically significant difference.

## Discussion

Risk factors INR elevation in patients using CMZ were liver disorder, nutritional risk, no-diabetes mellitus, and concomitant administration of warfarin. In addition, we presented the probability of elevated INR for each risk score in patients with these risk factors, which we believe will contribute to the safe use of CMZ.

Liver disorder, nutritional risk, and anticoagulant use have previously been reported as risk factors affecting bleeding and coagulopathy [11,12] The liver is the site of production of coagulation factors for hemostasis, and patients with liver disorder experience a decline in the ability to synthesize coagulation factors. In addition, impaired excretion of bile acids into the intestinal tract results in impaired absorption of vitamin K. These factors result in INR elevation [24]. In patients with nutritional risk, lower dietary vitamin K intake may have led to elevated INR due to reduced functional activity of vitamin K-dependent clotting factors [25]. In the regulation of blood coagulation Vitamin K is reduced to reduced vitamin K by vitamin K reductase during the synthesis of different vitamin K-dependent proteins such as prothrombin and factors VII, IX and X. Reduced vitamin K is the active cofactor required for γ-carboxylation of glutamate. During this reaction, reduced vitamin K is then converted to vitamin K epoxide, which is converted back to vitamin K by VKOR [9,10]. The γ-carboxylated glutamic acid groups of these proteins promote the coagulation cascade by binding with $Ca^{2+}$. Vitamin K plays a vital role in clotting activity [9]. Warfarin inhibits VKOR and causes INR elevation. The reason why warfarin was the factor most strongly associated with elevated INR may be that it has the same mechanism of action as CMZ [26] In this study, no-diabetes mellitus was first identified as a risk factor for coagulopathy by antibiotics containing the NMTT group. It has been reported that diabetes mellitus patients are more prone to thrombosis than those without diabetes mellitus due to factors such as increased platelet reactivity, increased coagulation factors, and impaired fibrinolysis [27].

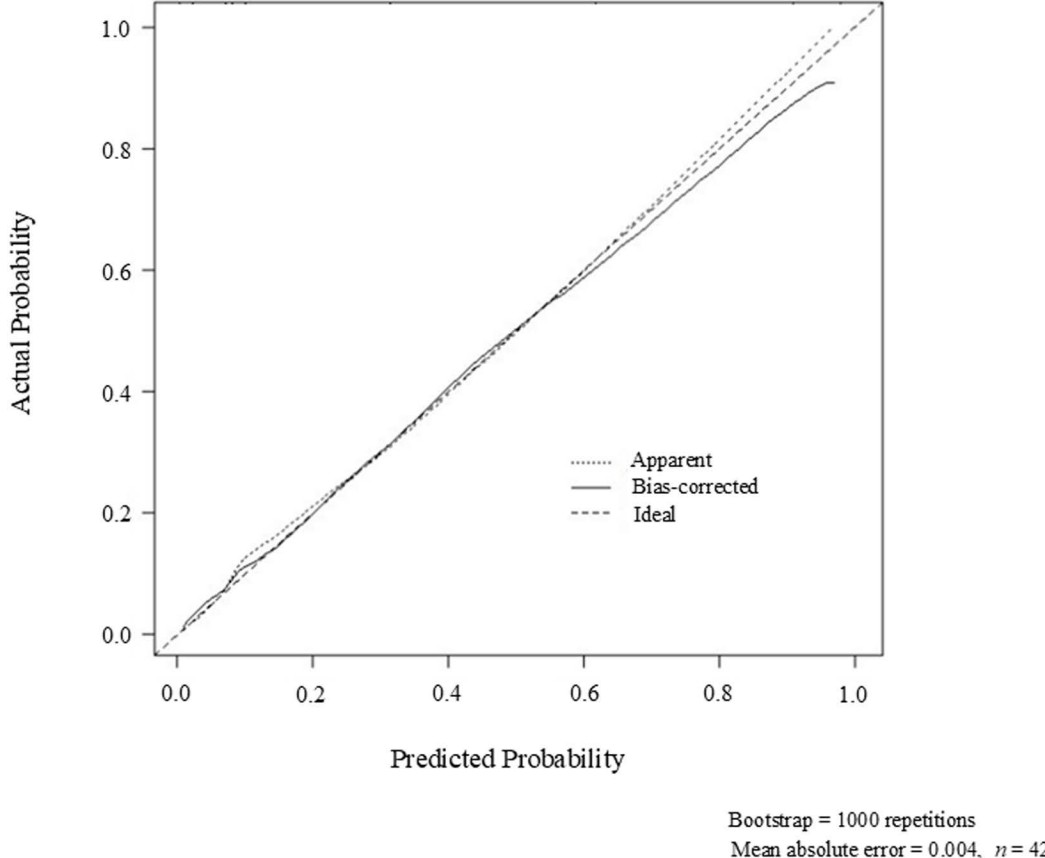

**Fig 3. Calibration curve showing consistency between the predicted probability (x-axis) with the observed real risk (y-axis) for elevated international normalized ratio (INR) associated with cefmetazole.** Perfect prediction is shown by the diagonal black dashed line. The black small, dotted line shows the entire cohort. The black solid line represents bias correction with bootstrapping.

It has also been reported that patients with diabetes mellitus require 1.5 to 2 times the dosage of warfarin to maintain INR as compared to patients without diabetes mellitus [28]. Therefore, the clotting tendency of diabetes mellitus patients may have suppressed INR elevation in patients using CMZ. It should be noted that the risk score for no-diabetes mellitus patients is 2 points, and with a risk score of 2, the risk of INR elevation is approximately 1%, so not all no-diabetes mellitus patients will develop coagulation disorders when using CMZ. However, we think it would be prudent to use CMZ while keeping in mind that the INR may elevate when combined with other factors such as liver disorder and nutritional risk.

The CL of unbound CMZ for each patient was calculated, and the AUC calculated using this was statistically significant at $P = 0.04$, but the aOR was 1.00 and had almost no effect on INR elevation. This might be owing to differences in blood collection dates and other correlation factors and was excluded from the final model. DOACs were dabigatran (0 in the INR-elevated group, 1 in the non-INR-elevated group), rivaroxaban (0 in the INR-elevated group, 5 in the non-INR-elevated), apixaban (5 in the INR-elevated group, 3 in the non-INR-elevated group), and edoxaban (2 in the INR-elevated group, 8 in the non-INR-elevated group). DOACs were statistically significant in univariate analysis, but not in multivariate analysis. If patients on CMZ require anticoagulants, using DOACs instead of warfarin may prevent INR elevation. However, since the effect on INR elevation varies depending on the type of DOACs, it is necessary to be careful

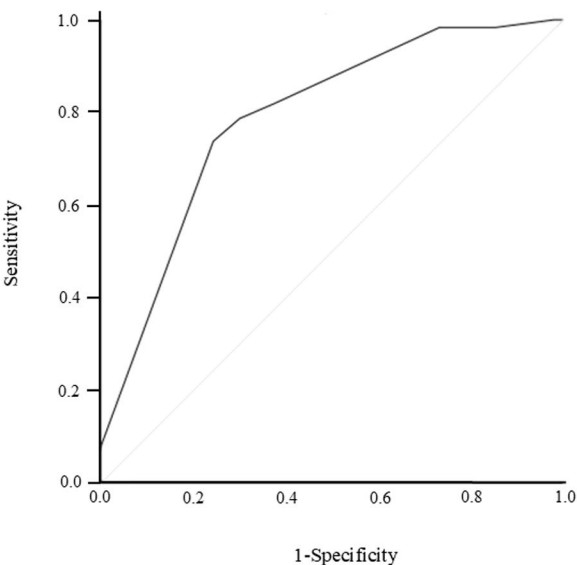

**Fig 4. Validation was performed using the receiver operating characteristic (ROC) curve for the discriminatory power of the predictive risk for the incidence of elevated international normalized ratio (INR) associated with cefmetazole.** Area under the curve, 0.79 (95% confidence interval, 0.73–0.85); $P < 0.001$; sensitivity, 78.7%; specificity, 69.8%.

when judging based on this result alone [29]. In this study, the factors identified as risk factors were not factors related to CMZ exposure, but factors affecting vitamin K and clotting factors. The synergistic effect of CMZ inhibition of VKOR and each risk factor suggests the possibility that coagulopathy may have appeared more strongly.

Patients taking warfarin are more likely to have their INR checked regularly. However, even in patients with stable INR prior to CMZ use, CMZ use was shown to result in elevated INR. Therefore, INR measurements should be considered for several days after starting CMZ. Patients not taking warfarin may not undergo regular INR measurements. However, patients with any one of these risk factors of liver disorder, nutritional risk, and no-diabetes mellitus were newly found to be at risk of elevated INR. In particular, patients possessing all of these risk factors are at high risk (probability of INR elevated: 30–<90%) and may require INR monitoring.

Previously, what kind of patients were likely to experience INR elevation among those using CMZ was unclear. However, it has become possible to calculate the risk score for each patient before using CMZ and use it after considering the probability of INR elevation. In addition, since the median maximum elevated INR after commencing the use of CMZ was observed on day 3, it is considered that elevated INR can be detected early by measuring INR around 3 days after starting CMZ. On the other hand, low-risk patients with a risk score of 0 or 2 had a very low probability of INR elevation. Low-risk patients are less likely to experience INR elevation and are expected to experience less bleeding. Therefore, it can be administered without excessive concern for bleeding-related side effects or frequent blood sampling for INR measurements. Due to the variety of patients in clinical practice, there is a need for a simple, visual tool to determine how much risk a patient is at risk for INR elevation before starting CMZ. Table 3 shows the probability of INR elevation in patients using CMZ for each risk factor possessed by the patient. Using this will enable the safe administration of CMZ. In addition, since there were no actual patients, validity evaluation was not possible, but when assuming a patient with the risk factor of warfarin use only (risk score 5), the probability of INR elevation was calculated by regression equation to be 20.9%. This probability of INR elevation is between the risk score of 4 and 6–7 points shown in Fig 2. In this study, the identified risk scores were 2 for three factors and 5 for one factor, and due to issues with the combination of explanatory variables in the regression equation constructed, the probability of INR elevation could not be evaluated for risk scores of

3 and 8. However, based on the above assumptions, it is assumed that the probability of INR elevation is between 2 and 4 for a risk score of 3, and between 6–7 and 9 for a risk score of 8.

Previous reports have shown that CMZ significantly increases bleeding [12]. There was a statistically significant difference when comparing bleeding after CMZ use between the INR-elevated group and the non-elevated group. In this study, elevated INR was defined as a ratio of a ratio of maximum INR to baseline INR (maximum INR divided with baseline INR), during CMZ use, greater than 1.25 [11]. Regarding the relationship between INR elevated and bleeding risk, a previous report reported that the higher the INR, the more bleeding occurred [13]. This criterion was used because the aim was to find patients with elevated INR at an early stage. Early use of vitamin K preparations should be considered if significant elevated is observed during INR monitoring to prevent bleeding.

A limitation of this study is that it is a retrospective study, so unobserved risk factors could not be identified. We were unable to collect VKOR gene polymorphisms due to study design limitations. Furthermore, as this study was based on data from one city hospital and one university hospital in Japan, the sample size was limited to 429 patients. In addition, because the study only included Japanese subjects, it is unclear whether similar results would be obtained in non-Japanese patients. Therefore, it is considered that the influence of factors other than the identified risk factors has not been comprehensively evaluated. For further investigations, it is necessary to expand the sample size by including more survey facilities and using big data.

## Conclusion

Liver disorder, nutritional risk, no-diabetes mellitus, and warfarin use were significantly associated with INR elevation in patients using CMZ. In addition, we successfully constructed a predictive model for INR elevation using risk scores. Even without warfarin use, INR should be monitored, as there is a high risk of INR elevation in patients with liver disorder, nutritional risk, and no-diabetes mellitus. CMZ can be used relatively safely in patients with no risk factors, or only one of the three risk factors other than warfarin. Therefore, in terms of safety, it has become possible to use the CMZ appropriately.

## Acknowledgments

The authors thank Hiromi Shikano and Izumi Nishisaka at the Division of Supporting Services, Tokyo Bay Urayasu Ichikawa Medical Center. The authors thank everyone at the Department of Hospital Information System Department, Keio University Hospital. We also thank Takanori Ogawa at Certara GK for advice regarding the analysis. We would like to thank Editage (www.editage.com) for English language editing.

## Author contributions

**Conceptualization:** Takaya Namiki, Yuta Yokoyama, Masaru Samura.

**Data curation:** Takaya Namiki, Motonori Kimura, Shogo Fukuda, Shoji Seyama, Osamu Iketani, Haruki Ishikawa.

**Formal analysis:** Takaya Namiki, Masaru Samura.

**Investigation:** Takaya Namiki.

**Methodology:** Takaya Namiki, Yuta Yokoyama.

**Supervision:** Tomonori Nakamura.

**Validation:** Takaya Namiki, Masaru Samura.

**Writing – original draft:** Takaya Namiki.

**Writing – review & editing:** Takaya Namiki, Yuta Yokoyama, Motonori Kimura, Shogo Fukuda, Shoji Seyama, Osamu Iketani, Masaru Samura, Aya Jibiki, Hitoshi Kawazoe, Hisakazu Ohtani, Naoki Hasegawa, Kazuaki Matsumoto, Hideki Hashi, Sayo Suzuki.

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
