## [Editor Report · Decision Letter 0]

PONE-D-24-16951Identification of Risk Factors and Development of a Predictive Model in Patients Using Cefmetazole for International Normalized Ratio ElevationPLOS ONE

Dear Dr. YOKOYAMA,

Thank you for submitting your manuscript to PLOS ONE. After careful consideration, we feel that it has merit but does not fully meet PLOS ONE’s publication criteria as it currently stands. Therefore, we invite you to submit a revised version of the manuscript that addresses the points raised during the review process.

1. Between 2012 and 2017- Rewrite this as it does not seem appropriate to start a paragraph with this.

2. Why were minors not included in this study?

3. IUPHAR/BPS - Kindly mention the fullform of this

4. Limit the discussion as it is very exhaustive and the deviating from the main reason of the study

5. Also elaborate regarding limitations of the study

Please submit your revised manuscript by Aug 31 2024 11:59PM. If you will need more time than this to complete your revisions, please reply to this message or contact the journal office at plosone@plos.org . Please include the following items when submitting your revised manuscript:

We look forward to receiving your revised manuscript.

Kind regards,

Girish Joseph, MD

Guest Editor

PLOS ONE

Journal Requirements:

2. Thank you for stating the following in the Competing Interests section: "Dr. H. K. received research funding from Eli Lilly outside of the submitted work. Dr. K. M. received grants from Meiji Seika Pharma, Sumitomo Pharma and Shionogi outside of the submitted work. T.N. received research funding from Chugai, Daiichi Sankyo, Kyowa Kirin, Otsuka Pharmaceutical, Sanofi, and Shionogi outside of the submitted work. Conflicts that the editors consider relevant to the manuscript content have been disclosed."

We note that you received funding from a commercial source: "Eli Lilly, Meiji Seika Pharma, Sumitomo Pharma and Shionogi, Chugai, Daiichi Sankyo, Kyowa Kirin, Otsuka Pharmaceutical,and Sanofi".

Within this Competing Interests Statement, please confirm that this does not alter your adherence to all PLOS ONE policies on sharing data and materials by including the following statement: ""This does not alter our adherence to PLOS ONE policies on sharing data and materials.” (as detailed online in our guide for authors http://journals.plos.org/plosone/s/competing-interests).  If there are restrictions on sharing of data and/or materials, please state these. Please note that we cannot proceed with consideration of your article until this information has been declared. 

3. In the online submission form, you indicated that "We do not make the data publicly available, but you can do so if you wish. Please contact the corresponding author."

Additional Editor Comments:

Between 2012 and 2017- Rewrite this as it does not seem appropriate to start a paragraph with this.

Why were minors not included in this study?

IUPHAR/BPS - Kindly mention the fullform of this

Limit the discussion as it is very exhaustive and the deviating from the main reason of the study

Also elaborate regarding limitations of the study

---

## [Author Response · Author response to Decision Letter 1]

9 Aug 2024

Responses to Reviewer #1

We deeply appreciate the reviewers' comments and suggestions. We have revised the manuscript as suggested and ensured that it adheres to the required format. Our point-by-point responses to reviewers’ comments are provided below. Furthermore, all the changes made in the revised manuscript are highlighted in yellow.

1) When submitting your revision, we need you to address these additional requirements. Please ensure that your manuscript meets PLOS ONE's style requirements, including those for file naming. The PLOS ONE style templates can be found at https://journals.plos.org/plosone/s/file?id=wjVg/PLOSOne_formatting_sample_main_body.pdf and

Thank you for your comment. Accordingly, we added the following statements: “Identification of risk factors and development of a predictive model in patients using cefmetazole for international normalized ratio elevation”, “Yuta Yokoyama1,3 *”, “* Corresponding author”, and “E-mail: yokoyama-yt@pha.keio.ac.jp (YY)”

Page 1 Lines 1–2, Page 1 Line 3, Page 1 Line 20, and Page 2 Line 22

All major sections were also modified to bold type, 18pt font and Sub-sections of major sections were modified to bold type, 16pt font.

Page 3 Line 26, Page 4 Line 47, Page 5 Line 70, Page 9 Line 140, Page 18 Line 221, Page 22 Line 300, Page 23 Line 307, and Page 24 Line 325 and Page 5 Line 71, Page 5 Line 90, Page 7 Line 118, Page 9 Line 141, Page 12 Line 168, Page 13 Line 182, and Page 17 Line 218

Furthermore, the abstract has been revised as required by the guidelines.

Pages 3 Lines 27–46

The position of tables and figures have also been revised in accordance with the guidelines.

Pages 10–11 Lines 150–167, Pages 15 Lines 211–214, Pages 16 Lines 215–217

Pages 9 Lines 148–149, Pages 13 Lines 196–198, Pages 14 Lines 203–206, Pages 14 Lines 208–210

Furthermore, we deleted " Address: 1-5-30 Shibakoen, Minato-ku, Tokyo 105-8512, Japan Tel.: +81-03-5400-2639; Fax: +81-03-5400-2651." and " Background, Methods, Results, Conclusions " and Author contributions.

Page 2 Lines 23–24 and Page 3 Lines 27, 33, 38, 44 and Page 23 Lines 314–324

2) Thank you for stating the following in the Competing Interests section: "Dr. H. K. received research funding from Eli Lilly outside of the submitted work. Dr. K. M. received grants from Meiji Seika Pharma, Sumitomo Pharma and Shionogi outside of the submitted work. T.N. received research funding from Chugai, Daiichi Sankyo, Kyowa Kirin, Otsuka Pharmaceutical, Sanofi, and Shionogi outside of the submitted work. Conflicts that the editors consider relevant to the manuscript content have been disclosed."

We note that you received funding from a commercial source: "Eli Lilly, Meiji Seika Pharma, Sumitomo Pharma and Shionogi, Chugai, Daiichi Sankyo, Kyowa Kirin, Otsuka Pharmaceutical,and Sanofi".

Within this Competing Interests Statement, please confirm that this does not alter your adherence to all PLOS ONE policies on sharing data and materials by including the following statement: ""This does not alter our adherence to PLOS ONE policies on sharing data and materials.” (as detailed online in our guide for authors http://journals.plos.org/plosone/s/competing-interests).

If there are restrictions on sharing of data and/or materials, please state these. Please note that we cannot proceed with consideration of your article until this information has been declared.

Thank you for your comment. Based on the comments, we have added the following to the cover letter: “Although Dr. H. K, Dr. K. M, and Dr. T. N have received funding outside of the research, this does not alter our adherence to PLOS ONE policies on sharing data and materials. We have submitted to ICMJE.”

Cover letter

3) In the online submission form, you indicated that "We do not make the data publicly available, but you can do so if you wish. Please contact the corresponding author."

Thank you for your suggestion. We have made the data available in a public repository. You can access the data through the following private link URL: https://figshare.com/s/a7cc966004fa34980f25

4) When completing the data availability statement of the submission form, you indicated that you will make your data available on acceptance. We strongly recommend all authors decide on a data sharing plan before acceptance, as the process can be lengthy and hold up publication timelines. Please note that, though access restrictions are acceptable now, your entire data will need to be made freely accessible if your manuscript is accepted for publication. This policy applies to all data except where public deposition would breach compliance with the protocol approved by your research ethics board. If you are unable to adhere to our open data policy, please kindly revise your statement to explain your reasoning and we will seek the editor's input on an exemption. Please be assured that, once you have provided your new statement, the assessment of your exemption will not hold up the peer review process.

Thank you for your comment. We have published the data as mentioned above. We request you to review it.

5) We note that you have included the phrase “data not shown” in your manuscript. Unfortunately, this does not meet our data sharing requirements. PLOS does not permit references to inaccessible data.

We require that authors provide all relevant data within the paper, Supporting Information files, or in an acceptable, public repository.

Please add a citation to support this phrase or upload the data that corresponds with these findings to a stable repository (such as Figshare or Dryad) and provide and URLs, DOIs, or accession numbers that may be used to access these data.

Or, if the data are not a core part of the research being presented in your study, we ask that you remove the phrase that refers to these data.

Thank you for your comment. The text “data not shown” has been deleted as the data are now available on Figshare. 

Responses to Reviewer #2

We really appreciate your time and effort in reviewing our manuscript and thank you for the insightful comments, constructive criticisms, and helpful advice. We have addressed the concerns raised by the reviewer and revised the manuscript to improve its overall readability.

1) Between 2012 and 2017- Rewrite this as it does not seem appropriate to start a paragraph with this.

Thank you for your comment. We have modified “Extended-spectrum β-lactamase-producing enterobacteria (ESBL-E) have become a major public health challenge owing to increasing number of patients affected by antimicrobial-resistant bacteria between 2012 and 2017, leading to increased morbidity and mortality rates [1,2].”

Page 4 Lines 48–50

2) Why were minors not included in this study?

Thank you for your confirmation. We excluded pediatric patients from the study owing to their increased vulnerability to vitamin K deficiency compared to adults, as well as the challenge of accurately diagnosing coagulopathy caused by CMZ in children. Various factors contribute to vitamin K deficiency in children, such as immature intestinal microflora and lower vitamin K production compared to adults, or low placental passage of vitamin K in neonates.

3) IUPHAR/BPS - Kindly mention the fullform of this

Thank you for your suggestion. Accordingly, we have modified “Drug and molecular target nomenclature conforms to the International Union of Basic and Clinical Pharmacology/British Pharmacological Society (IUPHAR/BPS) Guide standards”

Page 6 Lines 91–92

4) Limit the discussion as it is very exhaustive and the deviating from the main reason of the study

Thank you for your comment. As you pointed out, I have revised the discussion content and relevant references. The following sentence has been deleted.

“In this study, patients with an unstable baseline INR were excluded, and all patients with coagulopathy due to infections such as septic shock were excluded. Coagulopathy due to infection is caused by various factors, one of which is disseminated intravascular coagulation (DIC) [23]. D-dimer is generally used as an indicator of DIC [24], but at our institution it is not an item that is tested in general practice, and it is measured at the doctor's discretion if there is a concern about a coagulation abnormality. Since there was no difference between the two groups in patients for whom D-dimer was measured, it is thought that there was no coagulopathy due to infection.”

“Univariate analysis showed a statistically significant difference in the total dose of CMZ in INR-elevated group compared to non-INR-elevated group, but not a significant difference in daily dose. In the INR-elevated group, the median maximum INR during CMZ use was on day 3, compared to the INR non-elevated group, where the median maximum INR was observed on day 1. In this study, blood sampling was performed on arbitrary dates. The lack of significant difference in daily dose may be due to the difference in blood sampling date. The CMZ treatment period (period from the start of CMZ to the day of maximum INR) is also thought to be greatly affected by differences in blood sampling date. Therefore, the CMZ treatment period was not included in this analysis.”

“The reason for this may be due to the difference in blood sampling dates described above; the INR-elevated group received a higher dose, which may have affected the statistical significance of the AUC. In addition, AUC was considered to be the effect of other correlated factors and was excluded from the final model.”

“However, the number of patients who underwent bleeding was very small: 3 (4.9%) in the INR-elevated group and 3 (0.82%) in the INR-non-elevated group, so further research is considered necessary.”

“This means that the error in the INR measurement reagent is approximately 10%, and the increase is due to some factor other than measurement error (Manufacturer data reference).”

“23. Levi M, Keller TT, Van Gorp E, Ten Cate H. Infection and inflammation and the coagulation system. Cardiovascular Research. 2003. pp. 26–39. doi:10.1016/S0008-6363(02)00857-X”

“24. Adam SS, Key NS, Greenberg CS. D-dimer antigen: Current concepts and future prospects. Blood. American Society of Hematology; 2009. pp. 2878–2887. doi:10.1182/blood-2008-06-165845”

Page 18 Lines 225–230, Page 19 Lines 248–254, Page 19 Lines 256–258, Page 21 Lines 285–287, Page 21 Lines 288–290, Page 25 Lines 385–386, Page 25 Lines 387–388

Furthermore, the following sentence has been added to supplement the discussion.” This might be owing to differences in blood collection dates and other correlation factors and was excluded from the final model.”

Page 19 Lines 258–259

5) Also elaborate regarding limitations of the study

Thank you for your comment. As you pointed out, we have added the following sentence regarding the limitations of the study. “Furthermore, as this study was based on data from one city hospital and one university hospital in Japan, the sample size was limited to 429 patients. Therefore, it is considered that the influence of factors other than the identified risk factors has not been comprehensively evaluated. For further investigations, it is necessary to expand the sample size by including more survey facilities and using big data.”

Page 21 Lines 296–299

In addition, the following sentence has been deleted. “Further research is expected in the future for factors that have not been identified.”

Page 21 Lines 295–296 

The error is found in Page 3 Line 45.

We deleted “,”

Finally, we would like to express our gratitude for your comments and suggestions, which greatly helped us improve our manuscr

---

## [Decision Letter · Decision Letter 1]

PONE-D-24-16951R1Identification of risk factors and development of a predictive model in patients using cefmetazole for international normalized ratio elevationPLOS ONE

Dear Dr. YOKOYAMA,

Thank you for submitting your manuscript to PLOS ONE. After careful consideration, we feel that it has merit but does not fully meet PLOS ONE’s publication criteria as it currently stands. Therefore, we invite you to submit a revised version of the manuscript that addresses the points raised during the review process. Please accept our apologies for the delays in the peer review of this manuscript; it has been difficult to secure reviewers for this work. We have only been able to secure one review, but I am sending you their comments now to avoid further delays. You can find their comments attached to this email. Please do carefully consider their comments and respond to each point in your response to reviewers. 

We look forward to receiving your revised manuscript.

Kind regards,

Sarah Jose, Ph.D.

Staff Editor

PLOS ONE

Reviewers' comments:

Reviewer's Responses to Questions

**Comments to the Author**

1. If the authors have adequately addressed your comments raised in a previous round of review and you feel that this manuscript is now acceptable for publication, you may indicate that here to bypass the “Comments to the Author” section, enter your conflict of interest statement in the “Confidential to Editor” section, and submit your "Accept" recommendation.

Reviewer #1: All comments have been addressed

2. Is the manuscript technically sound, and do the data support the conclusions?

Reviewer #1: Yes

3. Has the statistical analysis been performed appropriately and rigorously? 

Reviewer #1: Yes

4. Have the authors made all data underlying the findings in their manuscript fully available?

Reviewer #1: Yes

5. Is the manuscript presented in an intelligible fashion and written in standard English?

Reviewer #1: Yes

6. Review Comments to the Author

Reviewer #1: (No Response)

7. PLOS authors have the option to publish the peer review history of their article (what does this mean? ). If published, this will include your full peer review and any attached files.

**Do you want your identity to be public for this peer review?** For information about this choice, including consent withdrawal, please see our Privacy Policy .

Reviewer #1: **Yes: ** Komsing Methavigul

---

## [Author Response · Author response to Decision Letter 2]

1 May 2025

Responses to Reviewer #1

We deeply appreciate the reviewers' comments and suggestions regarding our manuscript. Our point-by-point responses to reviewers’ comments are provided below. Furthermore, all changes made to the revised manuscript are highlighted in yellow.

Major comments

1) In abstract section, what group do you classify in patients with the predicted incidence probabilities of INR elevation ranging > 10 to < 30% and > 40 to 90%?

Thank you for pointing this out. We have defined the probability of INR elevation based on the total score of each risk factor, referring to previous reports, so that we can classify all occurrence probabilities. We classified the probability of INR elevation into four categories: low risk (< 5%), medium risk (5–< 30%), high risk (30–< 90%), and very high risk (≥ 90%). The following passage was added to the method section:

Page 8 Lines 129–131, Reference 23

“From the total score of each risk factor, four risk classifications of low risk (< 5%), medium risk (5–< 30%), high risk (30–< 90%), and very high risk (≥ 90%) were performed for each elevated INR occurrence probability[21,23]”

Furthermore, we revised the description in each related section.

Page 3 Lines 37–38

“The predicted incidence probabilities of INR elevation were < 5% (low risk), 5–< 30% (medium risk), 30–< 90% (high risk), and ≥ 90% (very high risk). ”,

Page 13 Lines 182–184

“Risk scores between 0 and 2 were low risk (predicted probability < 5%), a risk score of 4 was medium risk (predicted probability 5–< 30%), those between 6–7 were high risk (predicted probability 30–< 90%), and those of 9 or higher were very high risk (predicted probability of occurrence ≥ 90%).”

Page 19 Lines 256–257

“In particular, patients possessing all of these risk factors are at high risk (probability of INR elevated: 30–< 90%) and may require INR monitoring.”

Table 3 has also been accordingly modified.

2) The definition of elevated INR in data collection section (93th – 94th line and 266th line) should be a ratio of maximum INR to baseline INR (maximum INR divided with baseline INR).

Thank you for pointing this out. We corrected the notation as follows:

Page 6 Lines 93–95

“Elevated INR was defined as a ratio of maximum INR to baseline INR (maximum INR divided with baseline INR) during CMZ use greater than 1.25.”

Page 20 Lines 276–278

“In this study, elevated INR was defined as a ratio of a ratio of maximum INR to baseline INR (maximum INR divided with baseline INR), during CMZ use, greater than 1.25.”

3) What is the full name of DOAC in data collection section (109th line)? If DOAC is direct oral anticoagulant, please revised the sentence “DOAC were patients with no dose changes for during and within 7 days prior to CMZ use.”

Thank you for your comment. In this study, direct oral anticoagulant (DOAC) was written as direct oral anticoagulants (DOACs) because there were multiple target drugs. We initially wrote ‘DOAC’ in the part you pointed out, but we will standardize it to DOACs to indicate multiple DOACs. The official name is written on lines 80-81, so we will correct this to DOACs.

Page 7 Lines 110–111

“DOACs were patients with no dose changes for during and within 7 days prior to CMZ use.”

In addition, there were also places in the discussion where the ‘DOAC’ was written, so we will correct it to ‘DOACs’.

Page 19 Lines 247–249

“However, since the effect on INR elevation varies depending on the type of DOAC, it is necessary to be careful when judging based on this result alone.”

4) Why does the p-value for the liver disorders in multivariate analysis in Table 1 reach 0.005, while their p-value in Table 2 reach only 0.05? Which is wrong?

Thank you for pointing that out. We checked the analysis data and corrected Table 2 because the p-value was 0.005.

Table 2 has been accordingly modified.

5) In INR-elevated prediction model construction and accuracy verification section, what group do you classify in patients with score of 3, 5, and 8?

Thank you for your comment. This time, the value of the β coefficient calculated from the results of the binomial logistic regression analysis was converted into a risk. As a result, warfarin use was scored as 5, and other factors were scored as 2. There were no patients with risk scores of 3, 5, and 8, in the case of 5, so the validity could not be evaluated, but it is possible to calculate from the regression equation assuming a patient with the risk factor of warfarin use only. When calculated under these conditions, the probability of INR elevation was estimated to be 20.9%. This probability of INR elevation is between the risk score of 4 and 6–7 points shown in Fig 2. Risk scores of 3 and 8 cannot be calculated as risk score values among the combinations of explanatory variables in the regression equation constructed this time, so the probability of INR elevation cannot be predicted. However, based on the above assumptions, it is assumed that the probability of occurrence is between 2 and 4 or 6–7 and 9 for risk scores of 3 and 8. Therefore, the following content has been included in the Discussion section:

Page 20 Lines 267–274

“In addition, since there were no actual patients, validity evaluation was not possible, but when assuming a patient with the risk factor of warfarin use only (risk score 5), the probability of INR elevation was calculated by regression equation to be 20.9%. This probability of INR elevation is between the risk score of 4 and 6–7 points shown in Fig 2. In this study, the identified risk scores were 2 for three factors and 5 for one factor, and due to issues with the combination of explanatory variables in the regression equation constructed, the probability of INR elevation could not be evaluated for risk scores of 3 and 8. However, based on the above assumptions, it is assumed that the probability of INR elevation is between 2 and 4 for a risk score of 3, and between 6–7 and 9 for a risk score of 8..”

6) Why does patients with no-diabetes being the risk factors for CMZ-induced INR elevation? Your discussion proposed the clotting tendency of diabetes mellitus patients may have suppressed INR elevation in patients using CMZ, but healthy person without diabetes mellitus should not have coagulopathy. Please clearly explain in your discussion section.

Thank you for pointing this out. As you pointed out, it is thought that coagulation disorders will almost never occur in all no-diabetes mellitus patients who use CMZ. It has been reported that diabetic mellitus patients are more susceptible to thrombosis than no-diabetes mellitus patients due to various factors, as described in the discussion. However, as a result of this risk score analysis, no-diabetes mellitus patients had a score of 2, and the probability of INR elevation in patients with a score of 2 is approximately 1%. A tendency was shown that the risk of INR elevation when combined with other risk factors such as liver disorder and nutritional risk. Therefore, we noted that not all no-diabetes mellitus patients develop coagulation disorders, and that caution is required when combined with other risk factors.

Page 19 Lines 237–240

“It should be noted that the risk score for no-diabetes mellitus patients is 2 points, and with a risk score of 2, the risk of INR elevation is approximately 1%, so not all no-diabetes mellitus patients will develop coagulation disorders when using CMZ. However, we think it would be prudent to use CMZ while keeping in mind that the INR may elevate when combined with other factors such as liver disorder and nutritional risk.”

7) Another limitation of your study is only Japanese patients were included, so this risk score model had have limited to generalizability in other races. Please add this limitation in your discussion section.

Thank you for pointing this out. As you have stated, this study only included Japanese patients, and it was not possible to study non-Japanese patients. For this reason, we have added the following passage to the section detailing the limitations of the study.

Page 21 Lines 284–286

“In addition, because the study only included Japanese subjects, it is unclear whether similar results would be obtained in non-Japanese patients.”

Minor comments

1) In patient information part of materials and methods section, the dates of access to electronic medical record data in 73th-74th line should be used with the name of months instead of the number of months. It helps to reduce the confusion between month/date/year and date/month/year for readers.

Thank you for pointing that out. We have corrected it to the name of months.

Page 5 Lines 73–75

“The dates of access to electronic medical record data were from June/1/2023 to July/31/2023, after obtaining approval from the ethics committee of each facility.”

2) Please add the full name of INR under Table 1.

Thank you for your comment. We included the full form of “INR, international normalized ratio” under Table 1.

Table 1 has been modified accordingly.

Finally, we would like to express our gratitude to the reviewers for their comments and suggestions, which greatly helped us improve our manuscript.

---

## [Decision Letter · Decision Letter 2]

Identification of risk factors and development of a predictive model in patients using cefmetazole for international normalized ratio elevation

PONE-D-24-16951R2

Dear Dr. YOKOYAMA,

We’re pleased to inform you that your manuscript has been judged scientifically suitable for publication and will be formally accepted for publication once it meets all outstanding technical requirements.

Kind regards,

Mehmet Baysal

Academic Editor

PLOS ONE

Additional Editor Comments (optional):

Reviewers' comments:

Reviewer's Responses to Questions

**Comments to the Author**

1. If the authors have adequately addressed your comments raised in a previous round of review and you feel that this manuscript is now acceptable for publication, you may indicate that here to bypass the “Comments to the Author” section, enter your conflict of interest statement in the “Confidential to Editor” section, and submit your "Accept" recommendation.

Reviewer #1: All comments have been addressed

2. Is the manuscript technically sound, and do the data support the conclusions?

Reviewer #1: Yes

3. Has the statistical analysis been performed appropriately and rigorously? 

Reviewer #1: Yes

4. Have the authors made all data underlying the findings in their manuscript fully available?

Reviewer #1: Yes

5. Is the manuscript presented in an intelligible fashion and written in standard English?

Reviewer #1: Yes

6. Review Comments to the Author

Reviewer #1: (No Response)

7. PLOS authors have the option to publish the peer review history of their article (what does this mean? ). If published, this will include your full peer review and any attached files.

**Do you want your identity to be public for this peer review?** For information about this choice, including consent withdrawal, please see our Privacy Policy .

Reviewer #1: **Yes: ** Komsing Methavigul

---

## [Editor Report · Acceptance letter]

PONE-D-24-16951R2

PLOS ONE

Dear Dr. Yokoyama,

I'm pleased to inform you that your manuscript has been deemed suitable for publication in PLOS ONE. Congratulations! Your manuscript is now being handed over to our production team.

Kind regards,

on behalf of

Dr. Mehmet Baysal

Academic Editor

PLOS ONE